# Long Term Prognostic Value of Contractile Reserve Assessed by Global Longitudinal Strain in Patients with Asymptomatic Severe Aortic Stenosis

**DOI:** 10.3390/jcm11030689

**Published:** 2022-01-28

**Authors:** Rosina Arbucci, Diego M. Lowenstein Haber, María Graciela Rousse, Ariel K. Saad, Liliana Martínez Golleti, Natalio Gastaldello, Miguel Amor, Cristian Caniggia, Pablo Merlo, Gustavo Zambrana, Marcela Galello, Esteban Clos, Vicente Mora, Jorge A. Lowenstein

**Affiliations:** 1Cardiodiagnosis Department, Investigaciones Médicas, Viamonte 1871, CABA, Buenos Aires 1056, Argentina; lowediego@hotmail.com (D.M.L.H.); gracielarousse@gmail.com (M.G.R.); arielsaad@gmail.com (A.K.S.); lilimartinez012@gmail.com (L.M.G.); ngastaldello@gmail.com (N.G.); miguelamor68@gmail.com (M.A.); dlowe@bioimagenes.com.ar (C.C.); pablommerlo@gmail.com (P.M.); gfzambrana@yahoo.com.ar (G.Z.); marcelagalello@gmail.com (M.G.); estebanclos@hotmail.com (E.C.); lowensteinjorge@hotmail.com (J.A.L.); 2Department of Cardiology, Hospital Universitario Dr. Peset, 46017 Valencia, Spain; vmoral@comv.es

**Keywords:** aortic stenosis, exercise echocardiography, strain rate imaging, aortic valve replacement

## Abstract

Background. Left ventricle (LV) global longitudinal strain (GLS) at rest has shown prognostic value in patients (pts) with severe aortic stenosis (SAS). Contractile reserve (CR) during exercise stress echo (ESE) estimated via GLS (CR-GLS) could better stratify the asymptomatic patients who could benefit from early intervention. Aims. To determine the long-term prognostic value of CR-GLS in patients with asymptomatic SAS with an ESE without inducible ischemia. Additionally, to compare the prognostic value of CR assessed via ejection fraction (CR-EF) and CR-GLS. Methods. In a prospective, single-center, observational study between 2013 and 2019, 101 pts with asymptomatic SAS and preserved left ventricular ejection fraction (LVEF) > 55% were enrolled. CR was considered present with an exercise-rest increase in LVEF (Simpson’s rule) ≥ 5 points and > 2 absolute points in GLS. Patients were assigned to 2 groups (G): G1: 56 patients with CR-GLS present; and G2: 45 patients CR-GLS absent. All patients were followed up. Results. G2 Patients were older, with lower exercise capability, less aortic valve area (AVA), a higher peak aortic gradient, and less LVEF (71.5% ± 5.9 vs. 66.8% ± 7.9; *p* = 0.002) and GLS (%) at exercise (G1: −22.2 ± 2.8 vs. G2: −18.45 ± 2.4; *p* = 0.001). During mean follow-up of 46.6 ± 3.4 months, events occurred in 45 pts., with higher incidence in G2 (G2 = 57.8% vs. G1 = 42.2%, *p* < 0.01). At Cox regression analysis, CR-GLS was an independent predictor of major cardiovascular events (HR: 1.98, 95% CI 1.09–3.58, *p* = 0.025). Event-free survival was lower for patients with CR-GLS absent (log rank test *p* = 0.022). CR-EF was not outcome predictive (log rank test *p* 0.095). Conclusions: In patients with asymptomatic SAS, the absence of CR-GLS during ESE is associated with worse prognosis. Additionally, CR-GLS was a better predictor of events than CR-EF.

## 1. Introduction

Aortic stenosis is one of the most frequent valve diseases, with a prevalence of up to 5% in individuals older than 65 years, and its frequency is rising due to the increase in the population’s life expectancy [1]. Currently, aortic valve replacement (AVR) is indicated in patients with symptoms attributable to their valve disease, either spontaneous or elicited by an exercise test, or when there is left ventricular (LV) systolic dysfunction estimated using an ejection fraction (EF) < 55% [2,3,4]. Occasionally, determining the presence of clinical symptoms could prove challenging, due to the population’s age, to limited functional capacity by other causes, or, as this is a disease that sets slowly, because of a progressive self-limitation of physical activity as an unconscious and gradual adaptation to dyspnea.

In severe aortic stenosis (SAS), there is pressure overload with LV concentric hypertrophy offsetting wall pressure, and hence, LVEF does not estimate the real myocardial contractile function [5]. Indeed, several studies have shown that patients with SAS with a LVEF between 50% and 60% have a worse prognosis [6].

To address the issues mentioned above, better and simpler options are being investigated for the assessment of myocardial function. Global longitudinal strain (GLS) due to speckle tracking allows the behavior of subendocardial fibers that are more susceptible to ischemic damage and interstitial collagen deposition to be fundamentally assessed and has been shown to be a good early marker of myocardial impairment, with less interobserver variability than the LVEF [7].

Previous studies have communicated that the increase of the mean gradient across the aortic valve and the increase of the pulmonary artery pressure during exercise could have prognostic value. However, other studies have not found the same results [8,9,10].

Contractile reserve (CR), defined as the ability of the myocardium to increase pump function in response to an inotropic stimulus, has proven its prognostic value in various clinical scenarios. Previous studies have shown that in many patients with AS, the CR is impaired, when assessed using tissue Doppler or speckle tracking echocardiography [11,12]. However, its usefulness as a prognostic marker has been less studied. Hence, the aims of the present study were to determine the long-term prognostic value of CR assessed via GLS (CR-GLS) in patients with asymptomatic SAS during ESE without inducible ischemia and, additionally, to compare if CR assessed using LVEF (CR-EF) was like that assessed using CR-GLS.

## 2. Methods

### 2.1. Study Population

This was a single-center, observational and prospective study, conducted between May 2013 and October 2019. All patients with SAS who underwent ESE were evaluated. Those with wall motion abnormalities or diminished LV systolic function (LVEF < 55%) in their baseline echocardiograms, cardiomyopathies, other significant valve disease, or history of valve replacement were excluded. A total of 125 patients were considered eligible for the study. Four patients were excluded because of inadequate image quality for strain analysis, and an additional 20 were excluded because of the occurrence of symptoms and/or ischemia determined via wall motion abnormalities during the test. A total of 101 out of 125 patients were included (Figure 1). Baseline demographics are displayed in Table 1.

The ethics committee of the Department of Education and Research of Investigaciones Médicas S.A. of CABA, Argentina, grants approved the study “Long term prognostic value of Contractile Reserve Assessed by Global Longitudinal Strain in Patients with Asymptomatic Severe Aortic Stenosis” from The Cardiac Diagnostic Department of the Institution, led by Dr. Jorge Lowenstein (APPROVAL ID = 13.12.07/2013, 10 December 2013). An informed consent was signed by all patients included in the study.

### 2.2. Resting Echocardiogram

Echocardiograms were performed with Vivid E9 or E95 (GE Healthcare) machines and a 5 MHz matrix transducer, and two-dimensional images were acquired at 60–70 frames/s. Measurement of the usual echocardiographic parameters was performed according to the guidelines of the American Society of Echocardiography (ASE) [13]. They included comprehensive measurements of the LVEF, which was accomplished using a semiautomated border detection method and aortic mean and peak gradients. The AVA was estimated through the continuity equation.

### 2.3. Exercise Stress Echocardiogram

All ESEs were performed with a Schiller™ supine cycloergometer. Patients exercised according to the protocol. A symptom-limited graded maximum bicycle exercise test was performed in the semisupine position on a tilt table. After an initial workload of 25 W maintained for 2 min, the workload was increased every 2 min by 25 W. A 12-lead ECG was monitored continuously, and blood pressure was measured at rest and every 2 min during exercise. If patients were on blockers, they were asked to stop their medication 24 h before the test. The other medications, if any, were left unchanged. Patients with an abnormal exercise test were excluded from the present study.

The following parameters were evaluated via echocardiography: wall motion index (WMI), E/e’ ratio, pulmonary artery systolic pressure (PASP), peak aortic velocity and peak aortic gradient. CR-GLS and CR-EF were analyzed (ESE protocol, modified from Picano et al.) [14] (Figure 2).

The stress test was interrupted if the heart rate limit appropriate for the age was reached, or with the occurrence of any of the following: symptoms (dyspnea, angina, or syncope), a decrease in systolic blood pressure (SBP), lack of appropriate rise in SBP (<20 mmHg), excessive increase in SBP (≥220 mm Hg), ST-segment depression >2 mm, or complex ventricular arrhythmia.

A stress test was positive for inducible ischemia when the onset of wall motion abnormalities during exercise was confirmed, accompanied or not by symptoms or ECG abnormalities.

As stated before, the presence of symptoms or inducible ischemia led to the exclusion of the referred patient.

### 2.4. Strain Assessment

Global longitudinal strain was analyzed from the apical four-chamber, two-chamber, and long axis views and was considered as the average of 16 segments at rest and peak exercise. The percentage that appears in the center of the bullseye represents the average of the 4 apical segments.

GLS analysis was performed during the test using the automatic functional images (AFI) tool. This technique consists of manually marking three endocardial points (two basal and one apical), and an algorithm automatically traces three lines that follow the endocardial, mesocardial, and epicardial borders [15]. For each measurement it was checked that the endocardial tracing was correctly tracked throughout the cardiac cycle, and manual corrections were done when necessary. A GLS(%) value ≥ −18 was considered normal, according to the results from prior studies in normal subjects performed at our laboratory [16]; similar values are referred to in the literature [17]. Conventionally, GLS is presented as a negative value (since it represents the myocardial shortening in the longitudinal direction). To reduce intraobserver variability, two different measurements were performed for each patient and subsequently averaged.

### 2.5. Assessment of Contractile Reserve by Global Longitudinal Strain and by Ejection Fraction

The ∆LVEF and ∆GLS were defined as the difference between peak exercise and rest values. CR was considered present with an exercise-rest increase in the LVEF (Simpson’s rule) ≥ 5 points and >2 absolute points in GLS [18], respectively (Figure 3).

### 2.6. Follow-Up

Our center is focused on diagnostic imaging without outpatient consultations or hospitalization, for which, unfortunately, patients only attend the stress exercise echo, and later, medical consultations are carried out in other centers such as SAVR or TAVR. Despite these considerations, all patients were followed up between March 2020 and July 2020 through telephone interviews conducted by trained healthcare personnel who were unaware of the results of the ESE. The patients were thoroughly questioned to assess the occurrence of events during follow-up.

### 2.7. Major Events and Outcomes

Valve replacement was defined as aortic valve intervention due the appearance or progression of symptoms secondary to the underlying disease defined according to the criteria of the treating physician. A major cardiovascular event was defined as acute myocardial infarction (AMI), stroke, or death (cardiovascular death and all-cause mortality).

Statistical Analysis: According to the distribution, continuous variables were described as mean and standard deviation or as medians and interquartile range and were analyzed using a *t* test (for parametric variables) or Mann-Whitney U test (for non-parametric variables). Categorical variables were presented as a percentage and compared using the X² or Fisher’s exact tests. To initially explore the association between clinical variables and incidence of major cardiovascular events, simple logistic regression analysis was performed. Variables with a *p* value < 0.10 were included in multiple logistic regression analysis to assess independent prognostic markers.

A receiver operating characteristic curve (ROC) and area under the curve (AUC) were created from the corresponding logistic regression. They were used to assess the diagnostic ability of GLS at rest and at exercise and CR-GLS and CR-EF to predict major cardiovascular events. The Youden index [19] was calculated to find the optimal cutoff value for the GLS at rest and exercise prognostic value. The best cutoff values were used to categorize the variables; to determine the incremental prognostic value of CR-GLS compared to CR-EF, during 46.6 ± 3.4 months mean follow up, Kaplan-Meier curves with a log rank test were performed. A logistic regression model was created to evaluate the significance of the variables in the model (Omnibus test), the goodness of fit of the data to the model (Hosmer-Lemeshow test), and especially the strength of association of the model (R2 of Nagelkerke). Subsequently, a survival analysis of the Cox model was created, and CR-EF and CR-GLS AUC were compared. Inter and intraobserver variability of CR-GLS were assessed using the intraclass correlation coefficient (ICC) in 20 randomly selected SAS patients. For all analysis, a *p*-value < 0.05 was considered statistically significant. All analyses were performed using Stata software, version 14 (StataCorp, College Station, Texas, USA) and SPSS, version 21 (Armonk, NY, USA).

## 3. Results

A total of 101 patients with asymptomatic SAS and preserved LVEF, who underwent ESE with a negative result for inducible ischemia, were included. Patients were assigned to two groups according to CR-GLS. Group 1 (G1) included patients in whom CR-GLS was present, and Group 2 (G2) included patients in whom CR-GLS was absent (Figure 3).

The patients’ demographic and clinical characteristics are described in Table 1. Patients in G2 were older (G2: 72.7 ± 8.5 years vs. G1: 66.5 ± 14.1 years, *p* = 0.014). There were no differences between groups in the prevalence of cardiovascular risk factors, cardiovascular history, or usual medication.

### 3.1. Baseline Echocardiography

Most of the patients had degenerative etiology (81 pts, 80.2%), with no differences according to CR-GLS presence. Bicuspid valve etiology was present in 19.8% (20 pts), with uneven distribution between the two groups (26.8% in G1, 11.1% in G2, *p* = 0.049).

No significant differences were found between both groups in LV mass, LVEF, GLS, E/e’, and pulmonary artery systolic pressure (PASP).

Patients in G2 presented a higher transaortic velocity, higher mean aortic gradient, and smaller AVA (Table 1).

### 3.2. Exercise Stress Echocardiography

All patients had a wall motion index (WMI) of 1 (normal) both at rest and with exercise. Exercise tolerance and heart rate during peak exercise were lower in G2 patients. There were no significant differences in velocities or medium aortic gradients achieved during peak exercise between the two groups.

During ESE, patients in G1 had higher EF values and higher GLS absolute scores compared to G2. A similar difference was seen in ΔLVEF and ΔGLS between both groups.

CR-EF was present in 50 patients in G1 (89.3%) and in 27 patients in G2 (60%). Assessment of diastolic function and PASP during exercise showed no differences between the groups (Table 2).

### 3.3. Follow-Up

Mean follow-up was 46.6 ± 3.4 months and was similar in both groups. Major events were reported in 45 patients: 12 deaths (9 cardiovascular deaths), 31 AVRs, 1 AMI, and 1 stroke. After analyzing the average time in which the patients underwent an AVR, we estimate an overall mean of 17.5 months (sd = 2.75). We found no difference between groups (CR-GLS absent 14 vs. CR-GLS present 22.5 *p* = 0.14). Events were more frequent in G2: 26 events (57.8%) vs. 19 events (33.9%) in G1; *p* = 0.017 (Table 3).

Discriminative power of GLS at rest (AUC 0.65 IC 95% 0.55–0.76, *p* = 0.007) and at exercise (AUC ROC 0.69 CI 95% 0.58–0.79, *p* = 0.001) were determined. The best cutoff point for GLS at rest was −18% (sensitivity 76%, specificity 38%) and for GLS at exercise −20% (sensitivity 75%, specificity 60%). Discriminative power of CR-GLS (AUC ROC 0.62, CI 95% 0.51–0.73, *p* = 0.04) impressed slightly better than that of CR-EF (AUC ROC 0.58 CI 95% 0.47–0.7, *p*= 0.14) (Figure 4).

The results of the variables with which the model was created are: significance (Omnibus test: *p* = 0.016), the goodness of fit of the data to the model (Hosmer-Lemeshow test, *p* < 0.001) and especially the strength of association of the model (Nagelkerke’s R2 *p* = 0.07). Clinical and echocardiographic variables associated with major cardiovascular events in univariate and multivariate analysis are shown in Table 4. In survival analysis, only the absence of CR-GLS was independently associated with major cardiovascular events (HR 1.97 CI 95% 1.09–3.58, *p* = 0.025) vs. CR-EF (HR 1.69 CI 0.90–3.14, *p* = 0.09).

Kaplan-Meier analysis demonstrated that patients with absent CR-GLS had worse prognosis than those with CR-GLS present (log rank yest = 0.022) (Figure 5), while CR-EF could not differentiate prognosis (log rank test = 0.095) (Figure 6).

### 3.4. Intra- and Inter-Observer Variability of CR-GLS

Twenty patients were randomly selected to verify the intra- and inter-observer variability of GLS-CR (0.99 and 0.98 respectively). Both intra- and inter-observer ICCs revealed good repeatability for contractile reserve using GLS.

## 4. Discussion

Aortic stenosis is the most common valve disease requiring surgery in the developed world [20,21]. It is a complex, multifaceted and systemic disease, which is not restricted to the aortic valve, but rather entails a decrease in arterial distensibility and changes in LV geometry and function. When symptoms develop, in the malignant phase of the disease, AVR results in a clear reduction in mortality, hence, it is a class I indication [2,3,4]. However, management of true asymptomatic patients remains controversial. The conservative expectant management may result in late AVR, when myocardial impairment is already irreversible. Consequently, in the absence of definitive evidence for decision-making in the case of asymptomatic patients, risk stratification requires other parameters to be considered. The newer diagnostic techniques contribute additional information, adding tools that allow the severity of the disease and its impact on the myocardium to be quantified.

During the progression of aortic stenosis, left ventricular wall thickness increases in order to diminish wall stress, in an attempt to compensate for pressure overload. Subsequently, due to myocardial imbalance between oxygen supply and demand, both interstitial fibrosis and cellular apoptosis increase, associated to various degrees of systolic dysfunction, with the resulting prognostic impact [5,22]. Nonetheless, EF remains within normal limits, since it is conditioned not only by myocardial contractility, but also by ventricular geometry. In the context of concentric LV hypertrophy, EF usually overestimates ventricular function, since it is a chamber-related index that mainly depends on radial fiber thickening. In this scenario, assessment of the longitudinal function of the myocardium, dependent on the innermost subendocardial fibers which are the most susceptible to ischemic microvascular damage, allows its dysfunction to be detected earlier [22,23,24,25]. Along those lines, myocardial abnormalities assessed via GLS have shown a very good correlation with the presence of delayed enhancement assessed via cardiac magnetic resonance imaging [26,27].

An exercise test is an excellent diagnostic tool to assess patients’ functional class and hemodynamic response, contributing valuable prognostic information with a good safety profile [28]. The introduction of ESE has added further information. Several parameters have been proposed that may improve the sensitivity to stratify patients with SAS, such as measurement of mean valve gradient (≥18–20 mmHg) during exercise, the presence of pulmonary hypertension during peak exercise (≥60 mmHg), and recently the measurement of systolic flow rate during exercise (<270 mL/s) [8,9,29]. However, other investigators have not reported the same results regarding the increase in mean transaortic gradient and/or peak pulmonary pressure at peak exercise, probably due to the diverse variables influencing such parameters during stress [10]. CR reflects the ability of the myocardium to respond to different stressors. An abnormal CR seems to be an early sign of LV systolic impairment, due to incipient involvement of myocardial fibers, and could be considered a sensitive sign of subclinical LV dysfunction.

Previous studies have reported that the resting GLS allows the prediction of events during follow-up when its value is <−15% [30]. However, the results of our study showed that the mean resting GLS values (−18.79% ± 2.12) were at the lower limit of the values considered normal. For this reason, we understand that evaluating the behavior of the CR-GLS was an important tool to unmask subclinical ventricular dysfunction. However, it is important to highlight that our population does not represent patients with severe aortic stenosis in advanced stages, in which patients are sicker, more symptomatic, and have greater valvular calcification, greater aortic velocity, and greater myocardial fibrosis, for which reason, the ROC curve analysis does not have the high sensitivity and specificity that we would expect as the disease progresses.

Studies have reported that GLS at rest allows the prediction of events during follow-up when its value is <15% [30]. Among the patients in our study, the mean GLS was −18.79% ± 2.12, at the low threshold of normal values. In our population, it was therefore important to unmask subclinical ventricular dysfunction with the assessment of CR-GLS.

Several studies have evaluated the behavior of CR during ESE. Van Pelt et al., have investigated the behavior of myocardial longitudinal fibers, analyzing the S wave of tissue Doppler during a treadmill ESE. They found that patients with moderate to SAS exhibited less increase in tissue velocity during exercise compared to control subjects. Such a finding was interpreted as an early sign of LV systolic dysfunction, and those patients also exhibited less exercise capacity, less increase in systolic blood pressure, and increased BNP levels [11]. Another study by Donal et al., showed that in patients with moderate to SAS, GLS at rest, the increase in mean aortic gradient with exercise, and the low increase in GLS with exercise were associated with an abnormal ESE. In this study, the correlation between changes in GLS and the EF were verified to be markedly less than in control subjects without aortic stenosis, thus reinforcing the notion that EF per se is not a good indicator of myocardial function in these patients. However, the authors did not assess whether this observation had any prognostic value [12]. More recently, Levy-Neuman et al., studied 75 patients with moderate and SAS with a treadmill ESE. During a follow-up of 34.5 ± 3.5 months, the mean basal aortic gradient and peak longitudinal strain of the basal segments (cutoff point: 17.99%) were the only independent predictors of cardiovascular events during follow-up, which occurred in 60% of patients. In that study, patients displayed lower GLS and larger AVA values than in our investigation, even though the authors included patients with moderate aortic stenosis [31]. Of note, in our experience, measurement of longitudinal strain in the basal segments during peak exercise is technically more difficult and probably entails greater variability and margin of error due to the elevated heart rate during an ESE. For this reason, we propose measuring GLS. In addition, when analyzing the number of AVRs during follow-up, certain differences were observed with respect to Levy’s study that could be the consequence of several factors. Our findings report a rate of indication for surgery of 30%, a lower value in comparison to another study performed by Levy-Neuman et al., (44%) [31]. This fact may be explained by external factors relating to the environment surrounding the system as well as the system itself and by internal factors relating to user behavior and motivation. In addition, some patients may face barriers for access for medical consultation, including distance, transportation costs, out-of pocket expenditures, among others. These factors and barriers limit the access for medical consultation as well as the late indication for surgery. As a consequence, cardiovascular mortality can rise in these patients. As shown in the results, our study compared to Levy’s, showed that patients with severe aortic stenosis have a high mortality in long-term follow-up (9/101 vs. 7/75 patients respectively). This may be explained by the fact that barriers in access to medical consultation delay the times in which the benefit of intervention would change the course of their disease.

Agnieszka K. Lech et al., evaluated exercise-induced changes in LV GLS in a group of 50 patients with asymptomatic SAS versus a control group of 21 healthy people. GLS values at rest in both groups were within normal limits but were significantly higher in the control group. There was an accurate increase in GLS during exercise in both groups, but it was smaller in the SAS group, indicating a preserved functional reserve of the LV myocardium but smaller than in healthy individuals. In this study, the average values of peak GLS in the SAS group were higher than in our study, probably due to the fact that the patients were very young, with more bicuspid valves and without concomitant diseases.

In our study we further evaluated LV systolic function and CR via the EF for comparison with GLS behavior. We could separate a group with impaired CR-SLG that had a worse prognosis during follow-up. Lech’s results may not show differences due to the small size of the study population, and they did not have a longitudinal follow-up to compare both groups. [32]

Since 2006, we have incorporated the assessment of myocardial strain using speckle-tracking echocardiography in multiple clinical scenarios into routine practice in our laboratory, as well as the assessment of CR during ESE.

In the present study, we investigated the prognostic implications of CR-GLS during an ESE without inducible ischemia in patients with asymptomatic SAS and preserved EF and observed that in patients with decreased CR-GLS, this parameter was an independent predictor of the composite endpoint of major cardiovascular events, among which AVR was the main event, and in Cox regression analysis, this parameter was the main independent variable that predicted long-term risk. The discriminative power of CR-GLS was slightly better than that of CR-EF. In addition, a cutoff value of 20% GLS during peak exercise predicted a higher risk of requiring AVR in patients during follow-up.

In our work, there are other considerations to be noted. In opposition to the study published by Lancellotti et al. [9] who showed a relation between peak PASP > 60 mmHg and cardiovascular events, we did not find such correlation. In agreement with our results, Goublaire et al. [10] found no relation with PASP during follow-up (14 ± 8 months).

### 4.1. Clinical Implications

In the present study, we demonstrate the prognostic value of a parameter with known usefulness, such as CR. In this case, the information obtained with a new technological tool such as strain assessed using speckle-tracking echocardiography proved to be better than that obtained by EF.

Routine use of this measurement during ESE tests in patients with asymptomatic SAS may add useful information for clinical decision-making.

### 4.2. Limitations

During the follow-up, the main cardiovascular event was AVR, so we tried to assure that the interventions were due to the appearance of symptoms and not due to the stress echocardiographic results. Another limitation was that the two groups did not have similar baseline characteristics, however, CR-GLS was the only independent predictor of cardiovascular events.

Our study was carried out with GE equipment for measurement of GLS; values may be different for other vendors.

In addition, as our medical center focused merely on diagnostic imaging, no further information regarding the VARC 2 criteria [33] for the TAVI implantation was collected in the patients, and thus, it was not possible to include this information in the formal analysis. Future research that merges the clinical information with the diagnosis information is needed to explore and analyze the role of the VARC 2 criteria in patients.

Our results should be confirmed in future studies, including multiple centers and larger patient populations.

## 5. Conclusions

In patients with asymptomatic SAS, the absence of CR-GLS during ESE is associated with worse prognosis. Additionally, CR-GLS was a better predictor of events than CR-EF.

## Figures and Tables

**Figure 1 jcm-11-00689-f001:**
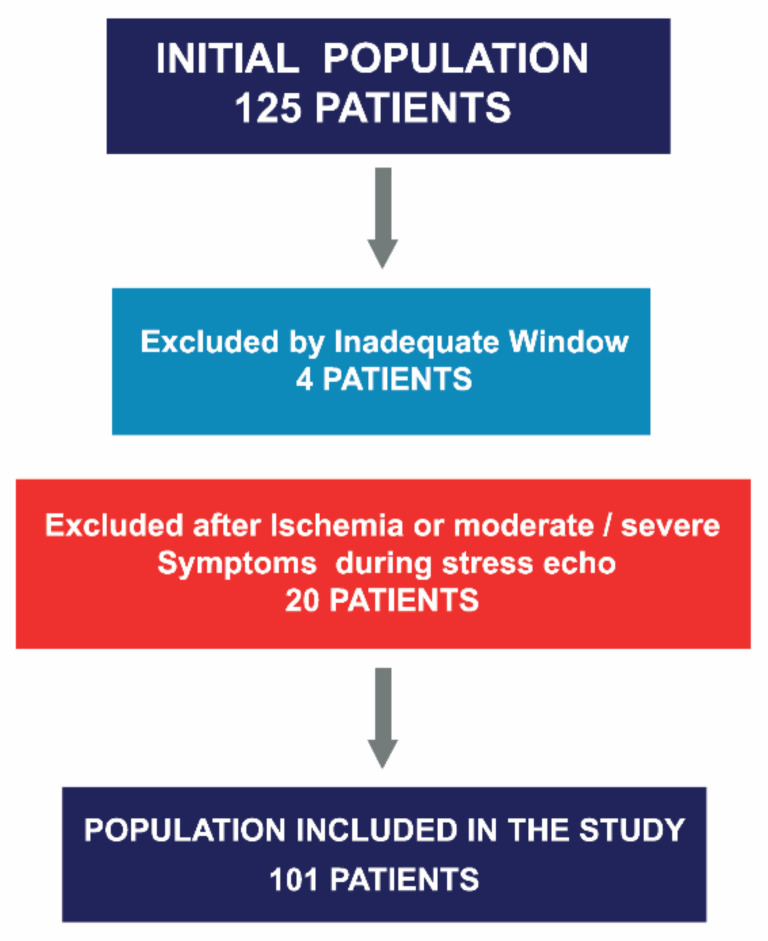
Asymptomatic severe aortic stenosis referred for stress echocardiography, patient selection.

**Figure 2 jcm-11-00689-f002:**
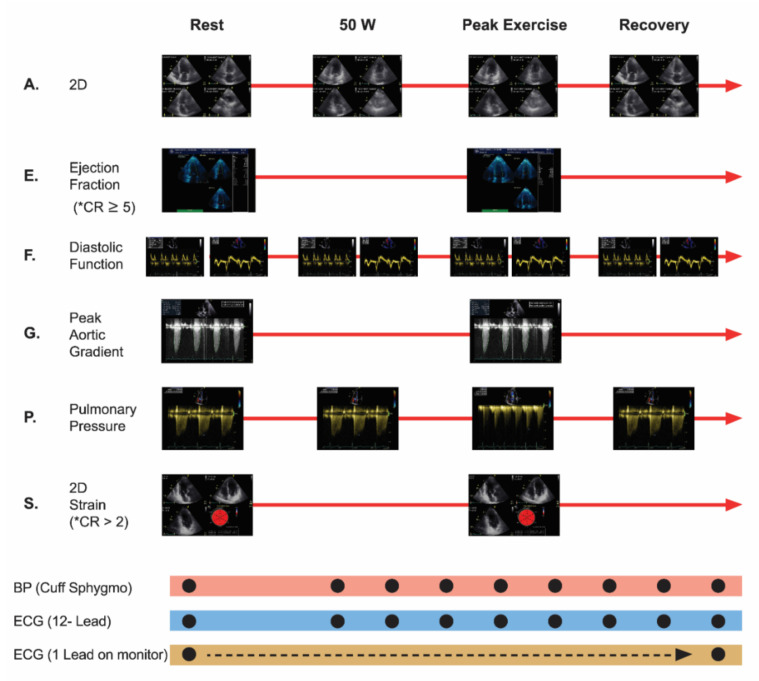
Exercise Stress Echocardiography Protocol, modified from Picano et al. New clinical standard of integrated quadruple stress echocardiography with ABCD protocol. Cardiovascular ultrasound 2018. 16:22. A: assynergies, E: ejection fraction, F: diastolic function, G: peak aortic gradient, P: pulmonary pressure, S: strain, * CR: contractile reserve was considered to be present when exercise to rest ejection fraction ratio measured via Simpson method was ≥5 absolute points, or >2 absolute points when measured via GLS. ECG: electrocardiogram.

**Figure 3 jcm-11-00689-f003:**
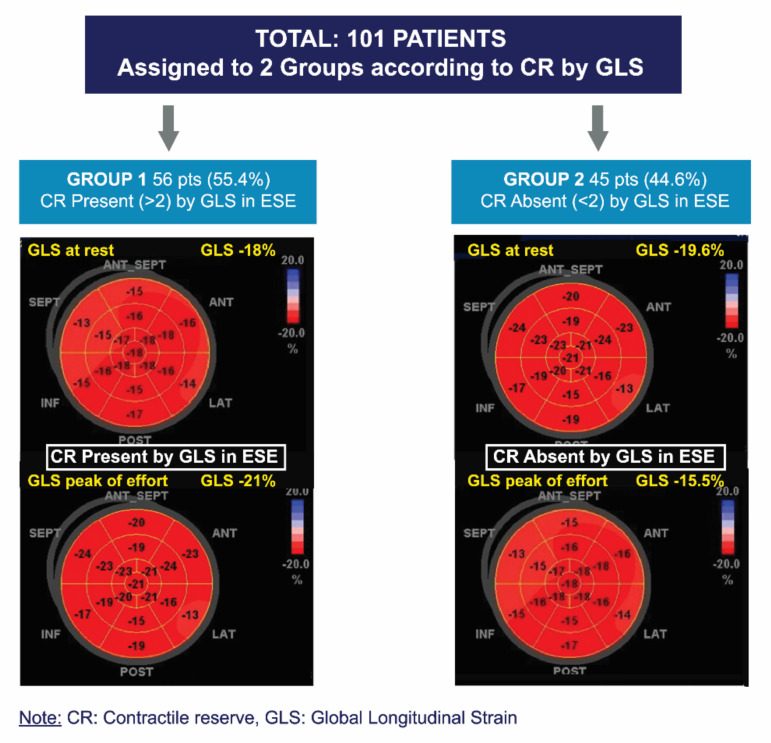
Division of groups according to presence or absence of contractile reserve via global longitudinal strain.

**Figure 4 jcm-11-00689-f004:**
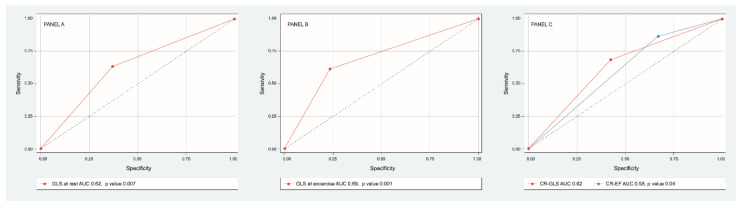
ROC curve of sensitivity and specificity of GLS at rest (panel **A**), GLS at exercise (panel **B**), CR-GLS vs. CR-EF (panel **C**) to predict major cardiovascular events.

**Figure 5 jcm-11-00689-f005:**
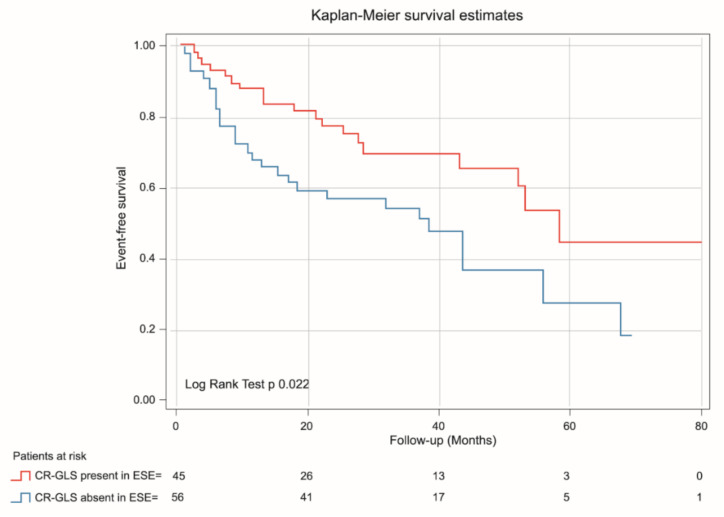
Kaplan-Meier estimates of event-free survival in asymptomatic SAS patients with CR-GLS during ESE present (red line) and CR-GLS during ESE absent (blue line). CR-SLG: contractile reserve assessed via global longitudinal strain, ESE: exercise stress echocardiography, SAS: severe aortic stenosis.

**Figure 6 jcm-11-00689-f006:**
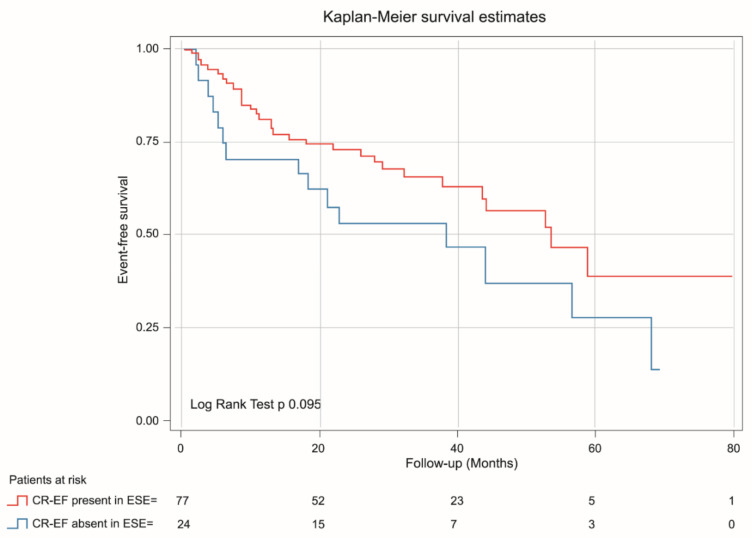
Kaplan-Meier estimates of event-free survival in asymptomatic SAS patients with CR-EF during ESE present (red line) and CR-EF during ESE absent (blue line).

**Table 1 jcm-11-00689-t001:** Clinical and echocardiographic variables at rest in both groups.

Variables	All Patients (*n* = 101)	G1: CR-GLS Present (*n* = 56 pts)	G2: CR-GLS Absent (*n* = 45 pts)	*p* Value
Age (years)	69.06 ± 12	66.54 ± 14.10	72.72 ± 8.52	0
Gender (male)	54 (53.5%)	33 (58.9%)	21 (46.7%)	0.21
HTN	61 (60.4%)	30 (53.6%)	31 (68.9%)	0.11
Prior MI without dyssynergies	3 (3%)	0 (0%)	3 (6.7%)	0.051
Prior PCI	2 (2%)	0 (0%)	2 (4.4%)	0.11
Prior CABG	10 (9.9%)	6 (10.7%)	4 (8.9%)	0.76
ACE Inh/ARB II	74 (73.3%)	41 (73.2%)	33 (73.3%)	0.98
Betablockers	40 (39.6%)	18 (32.1%)	22 (48.9%)	0.087
Calcium antagonists	40 (39.6%)	22 (39.3%)	18 (40%)	0.94
Statins	32 (31.7%)	17 (30.4%)	15 (33.3%)	0.74
SBP at rest (mmHg)	117.47 ± 6.18	117.2 ± 6.6	117.7 ± 5.59	0.66
HR at rest (bpm)	64 ± 6.15	64.6 ± 6.2	63.35 ± 6	0.29
Bicuspid Ao	20 (19.8%)	15 (26.8%)	5 (11.1%)	0.049
AVA (cm^2^) indexed at rest	0.46 ± 0.39	0.49 ± 0.03	0.44 ± 0.035	<0.001
LV Mass (g)	132 ± 19	133 ± 18.5	131.5 ± 19.8	0.67
Peak Ao velocity at rest (m/s)	4.58 ± 0.31	4.43 ± 0.28	4.78 ± 0.22	<0.001
Peak Ao gradient at rest (mmHg)	84. 54 ± 11.59	78.89 ± 10.6	91.58 ± 8.58	<0.001
Medium Ao gradient at rest (mmHg)	42.49 ± 5.9	39.8 ± 5.7	45.8 ± 4.3	<0.001
LVEF at rest (%)	63.23 ± 5.8	64 ± 5.6	62.1 ± 5.9	0.77
GLS at rest (%)	−18.79 ± 2.12	−19.14 ± 2.19	−18.37 ± 1.98	0.70
E/e’ at rest (cm/seg)	12.5 ± 4.7	12.3 ± 5	12.86 ± 4.5	0.60
PASP at rest (mmHg)	34.73 ± 7.5	34 ± 7.3	35.4 ± 7.8	0.40

Continuous variables are presented as mean ± SD. Categorical variables are expressed as number (percentage). HTN = hypertension, MI = myocardial infarction, PCI = percutaneous coronary interventions, CABG = coronary artery bypass grafting, ACE Inh = angiotensin converting enzyme inhibitors, ARB II = angiotensin II receptor blockers, AVA = aortic valve area, Ao = aortic, LVEF = ejection fraction, GLS = global longitudinal strain, CR = contractile reserve, PSP = pulmonary systolic pressure, FBPR = flat blood pressure response, SBP = systolic blood pressure, HR = heart rate, LV = left ventricle.

**Table 2 jcm-11-00689-t002:** Echocardiographic variables at exercise in both groups.

Variables	All Patients (*n* = 101)	G1: CR-GLS Present (*n* = 56 pts)	G2: CR-GLS Absent (*n* = 45 pts)	*p* Value
Peak Ao velocity exercise (m/s)	4.9 ± 0.36	4.88 ± 0.38	4.92 ± 0.34	0.57
Peak Ao gradient exercise (mmHg)	102.09 ± 19.2	100.7 ± 18	103.75 ± 20.5	0.43
Medium Ao gradient at exercise (mmHg)	51.03 ± 9.6	50.5 ± 9	52 ± 10	0.4
LVEF at peak exercise (%)	69.4 ± 7.22	71.5 ± 5.9	66.8 ± 7.9	0.002
GLS at peak exercise (%)	−20.53 ± 3	−22.2 ± 2.8	−18.45 ± 2.4	<0.001
E/e’ at peak exercise (cm/seg)	15.26 ± 2.9	14.8 ± 3.3	15.7 ± 2.5	0.12
PASP at peak exercise (mmHg)	50.71 ± 10.9	50.3 ± 11.8	51 ± 10	0.74
Δ LVEF (peak-rest)	6.16 ± 4.34	7.3 ± 2.9	4.7 ± 5.3	0.003
Δ Strain (peak-rest)	1.74 ± 2	3.07 ± 0.85	0.08 ± 1.9	<0.001
Present CR-EF (% pts)	77 (76.2)	50 (89.3)	27 (60)	0.001
Absent CR-EF (% pts)	24 (23.8)	6 (10.7)	18 (40)	0.001
METS	5.06 ± 1.7	5.6 ± 2	4.4 ± 1.1	0.002
WATT	92.16 ± 33.1	101.9 ± 35.5	80 ± 25	0.001
Medium Ao gradient at exercise (mmHg)	51.03 ± 9.6	50.5 ± 9	52 ± 10	0.4
SBP at peak exercise (mmHg)	168.94 ± 29.95	172.7 ± 32.6	164.2 ± 25.7	0.15
HR at peak exercise (bpm)	120 ± 23.3	125.5 ± 22.5	113 ± 22.6	0.008

Continuous variables are presented as mean ± SD. Categorical variables are expressed as number (percentage). LVEF = ejection fraction, GLS = global longitudinal strain, CR = contractile reserve, PASP = pulmonary systolic pressure, FBPR = flat blood pressure response, SBP = systolic blood pressure, HR = heart rate, LV = left ventricle.

**Table 3 jcm-11-00689-t003:** Cardiovascular events: long-term follow-up assessed in the different groups.

Events	All Patients (101)	G1: CR-GLS Present (*n* = 56 pts)	G2: CR-GLS Absent (*n* = 45 patients)	*p* Value
Combined	45 (44.6%)	19 (33.9%)	26 (57.8%)	0.017
Aortic Valve Replacement	31 (29.7%)	11 (19.6%)	20 (42.2%)	0.02
All-Cause Deaths	13 (12.9%)	6 (10.7%)	7 (15.6%)	NS *
Myocardial Infarction	1 (1%)	1 (1.8%)	0 (0%)	NS *
Stroke	1 (1%)	1 (1.8%)	0 (0%)	NS *

Note: NS * non-significant, CR: contractile reserve, CV: cardiovascular, GLS: global longitudinal strain.

**Table 4 jcm-11-00689-t004:** Univariate and multivariate predictor of major cardiovascular event.

	**Univariate Analysis**	
**Variable**	**HR (95% CI)**	** *p* ** **Value**
Age (years)	1.01 (0.98–1.049)	0.39
METS	0.73 (0.53–1.015)	0.06
KGM	0.99 (0.99–1.00)	0.23
Left ventricular mass (gr)	1.02 (0.99–1.04)	0.057
Bicuspid aortic valve	1.23 (0.46–3.4)	0.69
AVA indexed (cm2)	0.02 (0.00–5.3)	0.17
Medium Ao gradient at rest (mmHg)	1.06 (0.99–1.14)	0.10
Medium Ao gradient at exercise (mmHg)	1.01 (0.9–1.37)	0.15
PSP at rest (mmHg)	1.06 (0.99–1.13)	0.09
PSP at exercise (mmHg)	1.03 (0.99–1.07)	0.23
FBPR (%)	0.74 (0.31–1.8)	0.5
EF at rest (%)	0.99 (0.93–1.06)	0.8
EF at exercise (%)	0.96 (0.91–1.02)	0.18
GLS at rest (-%)	0.73 (0.58–0.91)	0.006
GLS at exercise (-%)	0.79 (0.68–0.92)	0.003
CR EF	2.61 (1.01–6.7)	0.049
CR GLS	2.66 (1.18–5.9)	0.018
	**Multivariate Analysis**	
**Variable**	**HR (95% CI)**	** *p* ** **Value**
METS	0.88 (0.56–1.22)	0.36
Left ventricular mass (gr)	1.03 (0.99–1.06)	0.96
PSP at rest (mmHg)	1.01 (0.94–1.08)	0.84
CR EF	1.69 (0.90–3.14)	0.09
CR GLS	1.97 (1.09–3.58)	0.025

## Data Availability

No data reported.

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
