# Peer review of "Long Term Prognostic Value of Contractile Reserve Assessed by Global Longitudinal Strain in Patients with Asymptomatic Severe Aortic Stenosis"

_jcm, 2022, doi:10.3390/jcm11030689_

Round 1
Reviewer 1 Report
Comments to the Author
The Authors present the study design entitled “Long term prognostic value of Contractile Reserve Assessed by Global Longitudinal Strain in Patients with Asymptomatic Severe Aortic Stenosis”.
This is a very interesting study, which could be helpful for all doctors in everyday clinical practice, especially considering the fact, that severity of aortal stenosis and qualification of the patients to the surgical treatment is still challenging, despite new imaging modalities and modern surgical techniques.
The reviewer appreciates the Authors’ effort and notices the important clinical value of the study.
Otherwise, I have some comments which need to be raised before my final decision:
Major comments:
- The Authors used a composite endpoint. Please clarify precisely your decision about that endpoint. MACE originally is another endpoint than that proposed by the Authors.
- The presented study would help doctors to qualify patients with severe AS and without symptoms for surgical treatment. Therefore, I strongly recommend performing additional statistics concerning another end-point: AVR as a single end-point.
- Only 30% of patients enrolled in the study (with severe AS) were finally qualified to AVR within a relatively long follow-up period. Please, raise that point in the discussion part and try to compare that results with data from the literature. Please, add the results about the average time from the study of the patients and AVR performing.
- The Authors present the results of logistic regression analysis. In the statistical part, the Authors mentioned logistic regression analysis and Cox hazard regression analysis. Please, clarify that in the statistical part and use Cox hazard regression analysis instead of logistic regression analysis as the stronger and more appropriate for that kind of study (with the follow-up period).
Minor comments:
- The Authors mentioned about 18 segments in STE analysis, but there are 17 in Bull eye’s presentation (some authors even use only 16 without apex)- please correct that information.
- In the methods part, there is a lack of information about B-lines, which are mentioned in the results part. Please, add the explanation to the Methods part.
Author Response
November 05, 2021
Rosina Arbucci, M.D. Ph.D.
Editor-in Chief
Cardiodiagnostic Investigaciones Médicas.
Manuscript ID: jcm-1428282
Type: Article
Title: Long term prognostic value of Contractile Reserve Assessed by Global Longitudinal Strain in Patients with Asymptomatic Severe Aortic Stenosis
We thank the reviewers and the Editor-in-Chief for their helpful comments and suggestions provided for our manuscript. All comments and reviews have been addressed. We list all comments and changes below in order to ease reviewers and editors' burden.
Rewiever number 1.
- The Authors used a composite endpoint. Please clarify precisely your decision about that endpoint. MACE originally is another endpoint than that proposed by the Authors.
Answer 1. Thanks for this comment. In the lines 141-145, we give more details about the Endpoint. The sentence now states: “Valve replacement and major cardiovascular event were defined as main events of the study. Valve replacement was defined as the presence of symptoms or progression of underlying disease defined according to the criteria of the treating physician that led to valve replacement. Major cardiovascular event s was defined as acute myocardial infarction (AMI), stroke or death (cardiovascular death and all-cause mortality)”.
- The presented study would help doctors to qualify patients with severe AS and without symptoms for surgical treatment. Therefore, I strongly recommend performing additional statistics concerning another end-point: AVR as a single end-point.
Answer 2. Thank you very much for your comment, we consider it very valuable to be able to incorporate it. Additional analyzes were performed with RVA as the single endpoint (Kaplan Meier and curve ROC).
Kaplan-Meier estimates of event-free survival for AVR in asymptomatic SAS patients with CR-EF during ESE present and CR-EF during ESE absent.
Receiver operating characteristic, curve (ROC) and area under the curve (AUC) were used to assess the diagnostic ability of GLS at rest and at exercise, CR-GLS and CR-EF to predict AVR. Discriminative power of GLS at rest (AUC 0.63 IC 95% 0.52-0.73, p=0.005) and at exercise (AUC ROC 0.69 CI 95% 0.57-0.77, p=0.005) were determined. Discriminative power of CR-GLS (AUC ROC 0.62, CI 95% 0.52-0.71) impressed slightly better than that of CR-EF (AUC ROC 0.58 CI 95% 0.48-0.68) p value= 0.005).
Additional graphics will be available as supplemental material to the article
Only 30% of patients enrolled in the study (with severe AS) were finally qualified to AVR within a relatively long follow-up period. Please, raise that point in the discussion part and try to compare that results with data from the literature. Please, add the results about the average time from the study of the patients and AVR performing.
Answer 3. Thank you very much for your comment, we consider your discussion very interesting. First of all, we want to clarify that our center is only for diagnostic imaging without outpatient clinics or hospitalization, for which, unfortunately, patients only attend to exercise stress echo and then, medical consultations are carried out in other centers as well how SAVR or TAVR. We will incorporate this point in the discussion section, lines 316 and 325, the sentence now states: “In addition, when analyzing the number of AVRs during follow-up, certain differences were observed with respect to Levy's study that could be the consequence of several factors. Our findings report a rate of indication for surgery of 30%, a lower value in comparison to other study performed by Levy-Neuman et al. (44%)31. This fact may be explained by external factors refer to the environment surrounding the system as well as the system itself, and by internal factors refer to user behavior and motivation. In addition, some patients may face barriers for access for medical consultation, including distance, transportation costs, out-of pocket expenditures, among others. This factors and barriers limit the access for medical consultation as well as the late indication for the surgery. As a consequence, the cardiovascular mortality can rise in this patients”
The Authors present the results of logistic regression analysis. In the statistical part, the Authors mentioned logistic regression analysis and Cox hazard regression analysis. Please, clarify that in the statistical part and use Cox hazard regression analysis instead of logistic regression analysis as the stronger and more appropriate for that kind of study (with the follow-up period).
Answer 4. Thank you very much for your comment, we consider it very valuable to be able to incorporate it. The modifications were made in the manuscript, line 221-225. In the line 226 the table was replaced.
Minor comments:
- The Authors mentioned about 18 segments in STE analysis, but there are 17 in Bull eye’s presentation (some authors even use only 16 without apex)- please correct that information.
Answer 1. Thank you very much for your comment, we consider it very valuable to be able to incorporate it. The modifications were made in the manuscript, line 117.
- In the methods part, there is a lack of information about B-lines, which are mentioned in the results part. Please, add the explanation to the Methods part.
Answer 2. Thank you very much for your comment, no additional comments were made on B lines because it was not one of the objectives of the study, we appreciate your comment and we believe it would be very useful to raise it as the objective of a new topic in future research.

Reviewer 2 Report
I read with interest the paper "Long term prognostic value of Contractile Reserve Assessed by Global Longitudinal Strain in Patients with Asymptomatic Severe Aortic Stenosis" by Rosina Arbucci et. al.
With their work, the authors try to devise a method to differentiate prognosis between patients with asymptomatic severe aortic stenosis (aSAS) using a combination of exercise stress echocardiography (ESE) with contractile reserve (CR) as assessed by global longitudinal strain (GLS; CR-GLS).
Using a model of first, simple (for individual parameters), and then multiple logistic regression (for those that had a p<0.10 in the former test) they found that only the parameter "absence of CR-GLS" showed a worse prognosis. The end point chosen was the first occurance of a major cardiovascular event.
The study covers a very interesting topic and the combination of CR with GLS is very intriguing. However, there are minor and major points that need to be addressed.
Major issues, general:
- The english language should be thoroughly re-ckecked. At this moment, it is a bit hard to read.
- There are many redundancies. Some are necessary, yes. Especially, I do get the importance of mentioning that patients that showed abnormalities in the ESE needed to be excluded. However, this should not be over-mentioned. In my opinion, a clear definition in the methods section and then in the discussion should be enough. But there are more examples to this point. Please check.
- Consitency: I understand, that it is somewhat difficult to describe the use of absolute change in percent in LVEF and GLS as a parameter. At this point, the definition is repeated multiple times - probably as a reminder (see "redundancy" above), but it also might confuse the reader. I suggest to also define this once in the methods section and then sticking to the mere values.
- Consistency: Please use the "- GLS %", i.e. "negative". This touches the same topic as with LVEF/GLS % above: at the moment, it is used interchangeably. Although this might be harder for the reader unversed in strain analysis, however, this is how it is done. So I suggest staying with the standard.
- Consistency: I suggest to design table #1 (baseline) according to table #2. Maybe, table #1 could start with the usual demographics, followed by "at-rest measurements". Those should follow the same order as those later displayed in table #2. Redundancy: "Peak Ao Velocity / Gradient exercise" is repeated in table #2 and hence can be deleted in table #1.
- Methods: The approval by an Institutional Research Committee might not be enough for all readers. Was also a vote from the local ethics committee obtained? This seems especially important, since this study is based on data prospectively collected.
- Methods: Also Zoghbis "Recommendations for Noninvasive Evaluation of Native Valvular Regurgitation", DOI: 10.1016/j.echo.2017.01.007 should be cited
- Methods: The proper Astrand protocol used should be cited: there are quite many modifications published, especially since the original - in my opinion - reflects treatmill, while Ii suspect a modification for bike was used here.
- Follow-up (FUP) should be described in more detail: when were the the FUP dates, and / or how long were the intervals. FUP was only done by telephone interviews, not on-site patient contacts. This should be briefly discussed.
- The endpoints are well described. However, they should be put in context with the VARC-2 criteria, including proper citation and also a discussion on why not all criteria were applied. Some, like "acute renal failure", might seem more obvious due to the type of FUP. However, I grievously miss bleeding, vascular complications, or at least hospitalisation - which, using the VARC-2 criteria, is connected to any of the parameters. Especially hospitalisation due to new heart failure is of high interest. This should also be at least discussed and, even better, re-assessed, since "Patients were questioned thoroughly to assess the occurrence of events during follow-up".
- The cardiovascular mortality seems quite high for a prospective study that followed up on patients at risk. First, it would be interesting to know if the study was carried out at the same place were TAVR and SAVR were or would have been performed. Second, this strongly underlines the risk these patients are confronted with. This point should be highlighted in the discussion, since we are talking about asymptomatic patients.
Major issues, statistics:
- With the use of regression, a model is created. In order to see if the model works, it seems important to see the significance of the variables / coefficients in the model (Omnibus test), the goodness-of-fit of the data to the model (Hosmer-Lemeshow test) and especially the strength of association of the mdoel (Nagelkerke's R2). This should be provided.
- Also, it is not clear how the ROC was created. I assume that it was derived from the corresponding logistic regression analyses.
- The AUC in the ROC curve is quite low, yielding cut-offs with only poor sensitivity and specificity. This is a major limitation and it should be deeply discussed, including the synthesis that the work can only be hypothesis generating at this stage.
- In the methods' section, it is stated that "Variables with p value <0.10 were included in multiple logistic regression analysis to assess independent prognostic markers.". This is somewhat confusing: first, what do you mean by "multiple logistic regression analysis"? Is this the use of the "significant variables" as independent variables in one logistic regression? The use of "multiple logistic regression" could also mean, that multiple logistic regressions were done with the same dependent but one single changing independent variable, but this is what was done before: "To initially explore the association between clinical variables and incidence of major cardiovascular events, simple logistic regression analysis was performed.", correct? If so, why were these variables chosen to explore? Finally, as already mentioned, the cutoff for an independent variable to be chosen in the "bigger" logistic regression analysis, was p<0.10. Why then, seems table 4 missing some of these variables under "Multivariate analysis" from the "Univariate analysis" section, i.e. "Medium Ao Gradient at rest" (not clear if 0.10 is e.g. "0.1044" or "0.0996" - in any case, clinically, it seems significant to me, as the baseline problem most likely will reflect on the outcome), GLS at rest (-%), GLS at exercise (-%), and CR EF?
- Please re-visit "all-cause death": in table 3, it seems to me, that "CV-death" (n=9, 9%) should be part of "all-cause death" (now with n=3, consider new n=9 + 3=12, 12 %). If calculated separately, as suggested by the table, this might need recalculation.
Minor issues:
- Abstract: It is unclear, which is which (LVEF or GLS): "...less LVEF and GLS at exercise (G1: 22.2%±2.8 vs. G2: 18.45%±2.4; p=0.001)".
- Methods: The sentence "After exclusion criteria were applied, a total of 101 patients were included as the study population (age: 69±12 years, 53.5% men) with a diagnosis of asymptomatic SAS (defined by Aortic Valve Area, AVA, <0.6 cm/m2 and/or medium gradient >40 mmHg according to current guidelines2-4) and preserved LVEF (>55%) (Figure 1)." contains much of data belonging to the results section. I suggest breaking this up and moving the hard data there, e.g. 101 of 125 patients were included (figure 1). Baseline demographics are displayd in table 1. etc.
- The sentence "Previous studies have communicated that the increase of the mean gradient across the aortic valve and the increase of the pulmonary artery pressure could have prognostic value." is not entirely clear. You mean "during exercise", correct?
- Methods: What is the type and origin (Swiss?) of the Schiller™ supine cycloergometer?
- Please check figure #3. According to my knowledge, the GLS should reflect the mean of all strain rates in the different segments. This seems not be the case in the examples given (compare DOI: 10.1186/s12947-019-0168-9; "GLS by speckle tracking echocardiography was measured
manually in a 18-segments LV model as the average segmental value based on three apical imaging planes"). Furthermore, in my copy, the quality of the images is a bit too low. - The use of "parametric" and / or "non-parametric" variables with the corresponding tests, instead of continuous, skewed continuous, categorial, etc. might be better.
- B-lines are only described in tale 4. However, this should be explained in the Methods' section and also in the results
Author Response
November 05, 2021
Rosina Arbucci, M.D. Ph.D.
Editor-in Chief
Cardiodiagnostic Investigaciones Médicas.
Manuscript ID: jcm-1428282
Type: Article
Title: Long term prognostic value of Contractile Reserve Assessed by Global Longitudinal Strain in Patients with Asymptomatic Severe Aortic Stenosis
Rewiever 2.
Major issues, general:
The english language should be thoroughly re-ckecked. At this moment, it is a bit hard to read.
Answer. Thanks for the comments made to the manuscript. We revised English style (done by a professional medical writer). We believe these changes clearly improved our manuscript.
There are many redundancies. Some are necessary, yes. Especially, I do get the importance of mentioning that patients that showed abnormalities in the ESE needed to be excluded. However, this should not be over-mentioned. In my opinion, a clear definition in the methods section and then in the discussion should be enough. But there are more examples to this point. Please check.
Answer. Thank you very much for your comment, we consider it very valuable to be able to incorporate it.
Consitency: I understand, that it is somewhat difficult to describe the use of absolute change in percent in LVEF and GLS as a parameter. At this point, the definition is repeated multiple times - probably as a reminder (see "redundancy" above), but it also might confuse the reader. I suggest to also define this once in the methods section and then sticking to the mere values.
Answer. Thank you for your comment, the modifications were made in the lines 143-145. The sentence now states: Contractile reserve was considered to be present when rest to exercise increase was ≥ 5 or > 2 LVEF percentage absolute points measured by Simpson or GLS20 respectively.
Consistency: Please use the "- GLS %", i.e. "negative". This touches the same topic as with LVEF/GLS % above: at the moment, it is used interchangeably. Although this might be harder for the reader unversed in strain analysis, however, this is how it is done. So I suggest staying with the standard.
Answer. Thank you for your comment, the modifications were made in the table and abstract.
Consistency: I suggest to design table #1 (baseline) according to table #2. Maybe, table #1 could start with the usual demographics, followed by "at-rest measurements". Those should follow the same order as those later displayed in table #2. Redundancy: "Peak Ao Velocity / Gradient exercise" is repeated in table #2 and hence can be deleted in table #1.
Answer. Thank you very much for your comment, we consider it very valuable to be able to incorporate it. In the line 180 and 201 the tables were replaced.
Methods: The approval by an Institutional Research Committee might not be enough for all readers. Was also a vote from the local ethics committee obtained? This seems especially important, since this study is based on data prospectively collected.
Answer. Thank you very much for your comment, below we attach the certificate of the ethics committee.
Methods: The proper Astrand protocol used should be cited: there are quite many modifications published, especially since the original - in my opinion - reflects treatmill, while Ii suspect a modification for bike was used here.
Answer. Thank you very much for your comment , changes were made to the lines 93-99.
Follow-up (FUP) should be described in more detail: when were the the FUP dates, and / or how long were the intervals. FUP was only done by telephone interviews, not on-site patient contacts. This should be briefly discussed.
Answer. Thank you very much for your comment, we consider it very valuable to be able to incorporate it. In the line 138-141. The sentence now states: “All patients were followed up between March 2020 and July 2020 by telephone interviews conducted by trained healthcare personnel who were unaware of the ESE results. Patients were questioned thoroughly to assess the occurrence of events during follow-up”.
The endpoints are well described. However, they should be put in context with the VARC-2 criteria, including proper citation and also a discussion on why not all criteria were applied. Some, like "acute renal failure", might seem more obvious due to the type of FUP. However, I grievously miss bleeding, vascular complications, or at least hospitalisation - which, using the VARC-2 criteria, is connected to any of the parameters. Especially hospitalisation due to new heart failure is of high interest. This should also be at least discussed and, even better, re-assessed, since "Patients were questioned thoroughly to assess the occurrence of events during follow-up".
Answer. Thank you for the recommendation. It would be interesting to perform the proposed analysis, we consider that this analysis exceed the objectives of the study, and that future studies could evaluate.
The cardiovascular mortality seems quite high for a prospective study that followed up on patients at risk. First, it would be interesting to know if the study was carried out at the same place were TAVR and SAVR were or would have been performed. Second, this strongly underlines the risk these patients are confronted with. This point should be highlighted in the discussion, since we are talking about asymptomatic patients.
Answer.
Thank you very much for your comment, we consider your discussion very interesting. First of all, we want to clarify that our center is only for diagnostic imaging without outpatient clinics or hospitalization, for which, unfortunately, patients only attend to exercise stress echo and then, medical consultations are carried out in other centers as well how SAVR or TAVR. We will incorporate this point in the discussion section, lines 316 and 325, the sentence now states: “In addition, when analyzing the number of AVRs during follow-up, certain differences were observed with respect to Levy's study that could be the consequence of several factors. Our findings report a rate of indication for surgery of 30%, a lower value in comparison to other study performed by Levy-Neuman and collegues (44%). This fact may be explained by external factors refer to the environment surrounding the system as well as the system itself, and by internal factors refer to user behavior and motivation. In addition, some patients may face barriers for access for medical consultation, including distance, transportation costs, out-of pocket expenditures, among others. This factors and barriers limit the access for medical consultation as well as the late indication for the surgery. As a consequence, the cardiovascular mortality can rise in this patients”
Major issues, statistics:
With the use of regression, a model is created. In order to see if the model works, it seems important to see the significance of the variables / coefficients in the model (Omnibus test), the goodness-of-fit of the data to the model (Hosmer-Lemeshow test) and especially the strength of association of the mdoel (Nagelkerke's R2). This should be provided.
Answer. Thank you for the recommendation. Additional tables will be available as supplemental material to the article
Also, it is not clear how the ROC was created. I assume that it was derived from the corresponding logistic regression analyses.
Answer. Thank you for the comments. The ROC was created from the corresponding logistic regression.
The AUC in the ROC curve is quite low, yielding cut-offs with only poor sensitivity and specificity. This is a major limitation and it should be deeply discussed, including the synthesis that the work can only be hypothesis generating at this stage.
Answer. Thank you for the comments, we understand that perhaps the low sensitivity and specificity of the ROC curve could be related to the fact that our population is not as ill as reported by previous studies in the world literature (for example, lancelotti, where patients have a baseline GLS of -15%, for which, they are sicker, with maoyr valvular calcification, greater aortic velocity, greater myocardial fibrosis.
Minor issues:
Abstract: It is unclear, which is which (LVEF or GLS): "...less LVEF and GLS at exercise (G1: 22.2%±2.8 vs. G2: 18.45%±2.4; p=0.001)".
Answer. Thank you for the comments, for a better understanding, the LVEF values were added in the line 23.
Answer.
Methods: The sentence "After exclusion criteria were applied, a total of 101 patients were included as the study population (age: 69±12 years, 53.5% men) with a diagnosis of asymptomatic SAS (defined by Aortic Valve Area, AVA, <0.6 cm/m2 and/or medium gradient >40 mmHg according to current guidelines2-4) and preserved LVEF (>55%) (Figure 1)." contains much of data belonging to the results section. I suggest breaking this up and moving the hard data there, e.g. 101 of 125 patients were included (figure 1). Baseline demographics are displayd in table 1. etc.
Answer. Thank you very much for your comment, we consider it very valuable to be able to incorporate it. In the line 74-75. The sentence now states: “101 of 125 patients were included (figure 1). Baseline demographics are displayd in table 1”.
The sentence "Previous studies have communicated that the increase of the mean gradient across the aortic valve and the increase of the pulmonary artery pressure could have prognostic value." is not entirely clear. You mean "during exercise", correct?.
Answer. Thank you very much for your comment, we refered to the exercise.
Methods: What is the type and origin (Swiss?) of the Schiller™ supine cycloergometer?
Answer. Thank you very much for your comment , Schiller is a Swiss stretcher, anyway, to give the reader a better understanding we decided to rewrite that part, in the line 93-99. The sentence now states: “Patients exercised according to the protocol A symptom-limited graded maximum bicycle exercise test was performed in the semisupine position on a tilt table. After an initial workload of 25 W maintained for 2 minutes, the workload was increased every 2 minutes by 25 W. A 12-lead ECG was monitored continuously, and blood pressure was measured at rest and every 2 minutes during exercise. If patients were on -blockers, they were asked to stop their medication 24 hours before the test. The other medications, if any, were left unchanged. Patients with an abnormal exercise test were excluded from the present study”.
Please check figure #3. According to my knowledge, the GLS should reflect the mean of all strain rates in the different segments. This seems not be the case in the examples given (compare DOI: 10.1186/s12947-019-0168-9; "GLS by speckle tracking echocardiography was measured manually in a 18-segments LV model as the average segmental value based on three apical imaging planes"). Furthermore, in my copy, the quality of the images is a bit too low.
Answer. Thank you very much for your comment, in figure 3: Global longitudinal strain was analyzed from 4, 3 and 2-chamber apical views and was considered as the average of 16 segments at rest and peak exercise. The porcentage that appears in the centre of the bull-eye represents the average of the 4 apical segments
The use of "parametric" and / or "non-parametric" variables with the corresponding tests, instead of continuous, skewed continuous, categorial, etc. might be better.
Answer. Thank you very much for your comment, parametric and non-parametric tests were used with continuous variables. Non-parametric tests were always used for categorical variables (example Chi2).
B-lines are only described in tale 4. However, this should be explained in the Methods' section and also in the results
Answer. Thank you very much for your comment, no additional comments were made on B lines because it was not one of the objectives of the study, we appreciate your comment and we believe it would be very useful to raise it as the objective of a new topic in future research. We understand that according to your proposal it is necessary to remove all kinds of information regarding B lines, therefore, I am going to remove it from all sections (with change control)

Round 2
Reviewer 1 Report
The Authors precisely corrected the text.
Author Response
No further comments were made by reviewer during the second round.
Reviewer 2 Report
I read the revision of "Long term prognostic value of Contractile Reserve Assessed by Global Longitudinal Strain in Patients with Asymptomatic Severe Aortic Stenosis" by Rosina Arbucci et al.
The authors have addressed some of my comments. However, there seems to be some misunderstanding and some things are missing.
The misunderstanding seems to stem from answering my comments / questions but not addressing the issue in the text. This should be done, however, because to the reader it is no use if reviewer and authors have an understanding, without letting the reader know.
So, relating to my original review, here are the remaining points:
- Consitency: I understand, that it is somewhat difficult to describe the use of absolute change in percent in LVEF and GLS as a parameter. At this point, the definition is repeated multiple times - probably as a reminder (see "redundancy" above), but it also might confuse the reader. I suggest to also define this once in the methods section and then sticking to the mere values.
- The definition of the label (- %) is confusing. So please use (%) and the values as -XX.
- Methods: The approval by an Institutional Research Committee might not be enough for all readers. Was also a vote from the local ethics committee obtained? This seems especially important, since this study is based on data prospectively collected.
Answer. Thank you very much for your comment, below we attach the certificate of the ethics committee.
- I am unable to find such a Certificate attachment. It is not in the supplementary file.
- This must be mentioned in the text that there was an ethic's committee vote with the appropriate reference number
- The endpoints are well described. However, they should be put in context with the VARC-2 criteria, including proper citation and also a discussion on why not all criteria were applied. Some, like "acute renal failure", might seem more obvious due to the type of FUP. However, I grievously miss bleeding, vascular complications, or at least hospitalisation - which, using the VARC-2 criteria, is connected to any of the parameters. Especially hospitalisation due to new heart failure is of high interest. This should also be at least discussed and, even better, re-assessed, since "Patients were questioned thoroughly to assess the occurrence of events during follow-up". Answer. Thank you for the recommendation. It would be interesting to perform the proposed analysis, we consider that this analysis exceed the objectives of the study, and that future studies could evaluate.
- I understand and expected as much. But please update the discussion. VARC-2 is the basis that should be used. If not, it should be discussed and added to the limitations.
-
The cardiovascular mortality seems quite high for a prospective study that followed up on patients at risk. First, it would be interesting to know if the study was carried out at the same place were TAVR and SAVR were or would have been performed. Second, this strongly underlines the risk these patients are confronted with. This point should be highlighted in the discussion, since we are talking about asymptomatic patients.
Answer: Thank you very much for your comment, we consider your discussion very interesting. First of all, we want to clarify that our center is only for diagnostic imaging without outpatient clinics or hospitalization, for which, unfortunately, patients only attend to exercise stress echo and then, medical consultations are carried out in other centers as well how SAVR or TAVR. We will incorporate this point in the discussion section, lines 316 and 325, the sentence now states: “In addition, when analyzing the number of AVRs during follow-up, certain differences were observed with respect to Levy's study that could be the consequence of several factors. Our findings report a rate of indication for surgery of 30%, a lower value in comparison to other study performed by Levy-Neuman and collegues (44%). This fact may be explained by external factors refer to the environment surrounding the system as well as the system itself, and by internal factors refer to user behavior and motivation. In addition, some patients may face barriers for access for medical consultation, including distance, transportation costs, out-of pocket expenditures, among others. This factors and barriers limit the access for medical consultation as well as the late indication for the surgery. As a consequence, the cardiovascular mortality can rise in this patients”
- Please also add in the methods section, that your center is focusing on diagnostics and that therapies are done elsewere. This helps understanding on how the work flow was in this study, e.g. that there was no direct access to FUP information, hence the telephone interviews.
- As mentioned, I strongly suggest to highlight in the discussion the mortality. This seems to me as a strong point, that could be used to emphasize, that asymptomatic patients with severe AS are not as "healthy" as they seem.
-
With the use of regression, a model is created. In order to see if the model works, it seems important to see the significance of the variables / coefficients in the model (Omnibus test), the goodness-of-fit of the data to the model (Hosmer-Lemeshow test) and especially the strength of association of the mdoel (Nagelkerke's R2). This should be provided. Answer. Thank you for the recommendation. Additional tables will be available as supplemental material to the article.
- I am unable to find supplemental material addressing the tests (It is not in the supplementary file). Furthermore, I think it would be better to mention the results of the tests in the text, since they belong to the regression analysis.
-
Also, it is not clear how the ROC curve was created. I assume that it was derived from the corresponding logistic regression analysis. Answer. Thank you for the comments. The ROC was created from the corresponding logistic regression.
- As subtly suggested by my comment, I do know that. But please also let the reader know.
-
The AUC in the ROC curve is quite low, yielding cut-offs with only poor sensitivity and specificity. This is a major limitation and it should be deeply discussed, including the synthesis that the work can only be hypothesis generating at this stage. Answer. Thank you for the comments, we understand that perhaps the low sensitivity and specificity of the ROC curve could be related to the fact that our population is not as ill as reported by previous studies in the world literature (for example, lancelotti, where patients have a baseline GLS of -15%, for which, they are sicker, with maoyr valvular calcification, greater aortic velocity, greater myocardial fibrosis.
- That might be the case. Please discuss it in the discussion section and let the reader know. Furthermore, this is a limitation, because the model seems very weak (hence the question on the other tests (s.a.)). Therefore, it seems to me that the study is "hypothesis generating".
- In the methods' section, it is stated that "Variables with p value <0.10 were included in multiple logistic regression analysis to assess independent prognostic markers.". This is somewhat confusing: first, what do you mean by "multiple logistic regression analysis"? Is this the use of the "significant variables" as independent variables in one logistic regression? The use of "multiple logistic regression" could also mean, that multiple logistic regressions were done with the same dependent but one single changing independent variable, but this is what was done before: "To initially explore the association between clinical variables and incidence of major cardiovascular events, simple logistic regression analysis was performed.", correct? If so, why were these variables chosen to explore? Finally, as already mentioned, the cutoff for an independent variable to be chosen in the "bigger" logistic regression analysis, was p<0.10. Why then, seems table 4 missing some of these variables under "Multivariate analysis" from the "Univariate analysis" section, i.e. "Medium Ao Gradient at rest" (not clear if 0.10 is e.g. "0.1044" or "0.0996" - in any case, clinically, it seems significant to me, as the baseline problem most likely will reflect on the outcome), GLS at rest (-%), GLS at exercise (-%), and CR EF?
- Unfortunately, this was not answered at all
- Please re-visit "all-cause death": in table 3, it seems to me, that "CV-death" (n=9, 9%) should be part of "all-cause death" (now with n=3, consider new n=9 + 3=12, 12 %). If calculated separately, as suggested by the table, this might need recalculation.
- Unfortunately, this was not answered at all
- The sentence "Previous studies have communicated that the increase of the mean gradient across the aortic valve and the increase of the pulmonary artery pressure could have prognostic value." is not entirely clear. You mean "during exercise", correct?
Answer. Thank you very much for your comment, we refered to the exercise.
- Again, I guessed as much. But please also let the reader know.
- The use of "parametric" and / or "non-parametric" variables with the corresponding tests, instead of continuous, skewed continuous, categorial, etc. might be better. Answer. Thank you very much for your comment, parametric and non-parametric tests were used with continuous variables. Non-parametric tests were always used for categorical variables (example Chi2).
- I only wanted to suggest to use the terms "parametric" and / or "non-parametric" instead of continuous, skewed continuous, categorial, in the text, because it is more precise. No problem.
Author Response
Dear reviewer find in the attachment point by point response to the comments
